# Digital Forensics System Using PLC for Inter-Floor Noise Measurement: Detailing PLC-Based Android Solution Replacing CCTV-based Solution

**Min-Ji Choo [1,2] and Jun-Ho Huh [3,\*]**

[1] IT Division, KODEHSHI AUK Group, Iksan 802-12, Korea; choo-minji@auk.co.kr or alicewww@etri.re.kr
[2] Electronics and Telecommunications Research Institute (ETRI), 218, Gajeong-ro, Yuseong-gu, Daejeon 34129, Korea
[3] Department of Data Informatics, Korea Maritime and Ocean University, Busan 600-716, Korea
[\*] Correspondence: 72networks@kmou.ac.kr; Tel.: +82-51-410-4371

**Abstract:** The neighborly dispute arising from the inter-floor noises has been increasing for the past two decades in Korean apartments or multi-unit houses sometimes leading to a serious consequence. Although there have been some attempts to resolve such a dispute, one of the underlying problems has to be solved first. That is, in addition to identifying the cause for the noise, it is necessary to prove who has actually suffered from the noise itself. Now that many of such a dispute is being settled at the civil court, producing objective evidence has become important. Therefore, digital forensics which is being widely used at the crime scenes by the police or other anti-crime organizations to collect evidences scientifically has begun to receive attention for the purpose of measuring noise levels. This technique has evolved in recent years following the rapid development in IT and ICT technologies but its problem is that such a technique has to be performed by the experts due to its complicated system so that those who need to measure the noise level usually outsource this work spending quite a sum. Thus, this study introduces a digital forensic system design which allows the user to effectively measure the inter-floor or neighborly noises directly and conveniently without much cost. The relevant system elements have been implemented with Java Android.

**Keywords:** digital forensics system; inter-floor noises evidence collection; PLC; ICT; black box; data collection; smart home

---

## 1. Introduction

Smart Homes are currently receiving attention from the IT industry. Various market forecasting organizations predict a rapid growth of these homes, and domestic and foreign companies are also launching many new services for them. Smart Homes also consider effective use of lighting, energy, and security devices, and they are connected with other smart systems such as smart city, smart education, smart key, etc. According to the "2013 Smart Home Industry Status Report" by the Korea Association of Smart Homes, the market size of the previous year (2012) was approximately 6.908 billion won, an 11.8% increase from the year before (6.161 billion won). The result of the survey targeting 156 companies associated with Smart Homes (2013) showed that they were positive about industries related to lighting, residential energy-saving devices/solutions, security imaging, and storage devices or Apps and peripherals for smart TVs [1–4]. Despite such various technological developments for Smart Homes, there are still many problems occurring at the residential houses in Republic of Korea (ROK) due to the noises generated between floors or adjacent units.

In ROK, residential noises such as inter-floor noise or environmental problems including the "right of light" are becoming a serious issue in our daily lives, causing excessive physical violence.

The number of cases involving environmental disputes is rapidly increasing and the statistics show that conflicts surrounding inter-floor and other residential noises have reached almost 85% of the total neighborhood disputes (as of the second half of 2016). Conflicts in the past were mostly related to factory-emitting smoke or river or atmospheric pollutions, since 1991, however, noise-related complaints have increased constantly.

During this period, complaints against atmospheric pollutions accounted for only 6% of the total complaints, but conflicts over solar access had increased instead. The total number of formal complaints made to the "Office of National Conflict Resolution Commission" in 2000 was 71, but the number increased more than three-fold to 215 (2014), whereas the noise-related ones numbered 25 cases in 2010 and also increased to 55 in 2014. Accordingly, the mental and material damages claimed by the victims against the perpetrators also increased from approximately 11.4% to 20.2% during the same period [5–7]. The main reason behind such an increase in the number of environmental disputes including inter-floor noise-related ones is that people have started to consider the mental damages, which they often used to take lightly, as serious physical damage that must be indemnified. For instance, Mr. A, who lives in an apartment house in Seoul, has been suffering from inter-floor noise for a year now. After hearing from the neighborhood center that it is possible to measure the noise, he contacted the center who asked about his specific situation. He replied that he was unable to sleep because of the noise occurring at dawn, sometimes until three A.M. A person from the center visited the family upstairs and asked them to keep the noise down after explaining the situation. In the meantime, the center asked Mr. A to keep a diary (prepared by the center) of noise occurrences after the mediation and send it by fax when the diary is completed to request noise measurement.

The family upstairs was quiet for the time being but soon became noisy again. Mr. A kept the diary until the last page, but he could not request noise measurement as the process takes about 24 h, and it was not easy for him to vacate his home. There are also times when his neighbor upstairs leaves their home or does not make any noise, so he could not easily set the day for the measurement. Mr. A said he was told to check the noise level (dB: unit of decibel) and see if it exceeds the limits every time he makes a complaint. Moreover, even if he attempts to check the noise level despite all the inconveniences, it will come to nothing if he cannot measure any noise while waiting for the right moment.

Among the typical microphones, a dynamic microphone was selected for this study as standard equipment. Meanwhile, the method used in the process of a receiver transforming and representing the sounds into decibels was devised by borrowing an idea from Fleming's Right-Hand Rule. In other words, the dynamic microphone operating as a receiver allows a sound wave (vibration) to be transformed into an electric signal through diaphragm vibration and we have used Fleming's Right-Hand Rule as a calculation formula.

Therefore, it is possible to build a more efficient smart home system by adopting a power line communication (PLC)-based digital forensic technology.

Thus, the proposed system design adopted the communications device and the Energy Storage System (ESS) on the power line, which is expected to be used for inter-floor noise forensics and as a household black box.

## 2. Related Research

In an attempt to reduce neighborly disputes arising from the inter-floor noises, the Ministry of Environment had set up the 'Neighborhood Center' under the management by Korea Environment Corporation in 2012. The center's role was to receive the noise-related complaints, conduct investigation by actually measuring the noise level for assessment, and then try to mediate between involved parties. Despite such an effort, it was pointed out that the center was unable to function properly as they could not deal with the ever-increasing number of complaints or respond to them in an effective way [5–8].

The center has been providing counseling to the complainants face-to-face or through online or phone calls for what actions they could take or try to intervene between disputing neighbors

before they seek any legal means. If this does work, they can also ask the Environmental Conflict Commission to assess the situation which involves making a judgment on the noise level measured by either side, focusing on the objectivity of the measurement or the method used. As mentioned earlier, measuring the noise level is not simple and cannot be achieved in a single attempt so that most of the victims tend to abandon any further legal pursuit and even if the noise has been measured correctly, the commission usually questions about its reliability or objectivity.

In such a case, the only other option is to seek outside professional help which can be costly compared to the free service offered by the neighborhood center. The companies or institutions specializing in noise measurement usually charge about 700,000 Korean won for their 24-h service.

Aside from its expensiveness, the cost can sometimes exceed the expected compensation: the compensation amounts were 520,000 Korean won, 663,000 Korean won, 793,000 Korean won, and 884,000 Korean won for the noises exceeding more than 5dB than legal limitations for the period of six months, one year, two years, and three years, respectively, all of which had not been quite satisfactory to the 41.9% of complainants in the past (Korea Environment Institution).

In response to such claims, the neighborhood center announced that, with the cooperation of the conflict resolution commission, their own measurement results will be used for the mediation, but the possibility of acceptance by the civil courts is uncertain. They also stressed that noise-related problems cannot be solved by simply measuring the noise levels, and that what is more important is to change or improve the life patterns of individual households so that they prioritize the counseling. Lastly, the center explained that, although people sometimes want to repeat measurement again when the results are not what they have expected, many others are waiting, so the same case cannot be dealt with again. Although the Korea Environmental Conflict Resolution Commission emphasizes residents' mutual concessions and considerations in the residential areas, the reality is that more attentive policies are needed to reduce noise-related disputes.

This study aims to assist victims of noise problems by proposing a system that can be used to collect objective evidence, support the filing of complaints, and facilitate a faster and more simple compensation process.

In the Republic of Korea, the number of inter-floor noise-related civil complaints received by the Neighborhood Inter-Floor Noise Complaint Center affiliated with the Ministry of Environment has increased from 8795 in 2012 to 28,231 in 2018. Although this organization is actively stepping in to mediate the dispute between neighbors, they do not seem to have any clear means of solving the problem as the causality involved in the noise-related issues are mostly due to the structural problems. The psychological damage can be compensated through a civil suit finding that the noises had exceeded the usual level one can tolerate but such a process requires a long judicial procedure.

Digital forensics is a new area of security service where the relevant fact of a certain activity can be investigated and proved based on digital data by using information equipment as a medium. It is being used for criminal investigations by the national investigation authorities such as police, prosecution, etc., and its necessity is increasingly recognized in the private sectors including financial or security companies as well. For example, this technic is quite useful in collecting legal evidence, preventing internal information, or strengthening internal security of auditing. Meanwhile, digital forensics is largely performed through the process of evidence collection, analysis, and submission. This study has focused on the evidence collection process.

### 2.1. Measuring Principle Using Microphone Sensors

Microphones are a device that converts sound signals into electrical signals, and telephone transmitters are one type of such device. Sound or noise is measured in decibels (dB) through the microphone sensor embedded in the mobile hardware.

Figure 1 shows the measuring principle of the microphone sensor. As in Figure 1, most Smart Phones usually have a microphone for talking at the bottom and another one at the top for shooting a picture or a video. The reason for having a separate video microphone is to maximize its serviceability

and sound quality. This microphone has another function, which is to maximize the quality of sound by removing the noises. As the method of this function, the distant sound is regarded as noise and eliminated by using the time difference between the sound entering the bottom microphone and the other sound entering the top microphone.

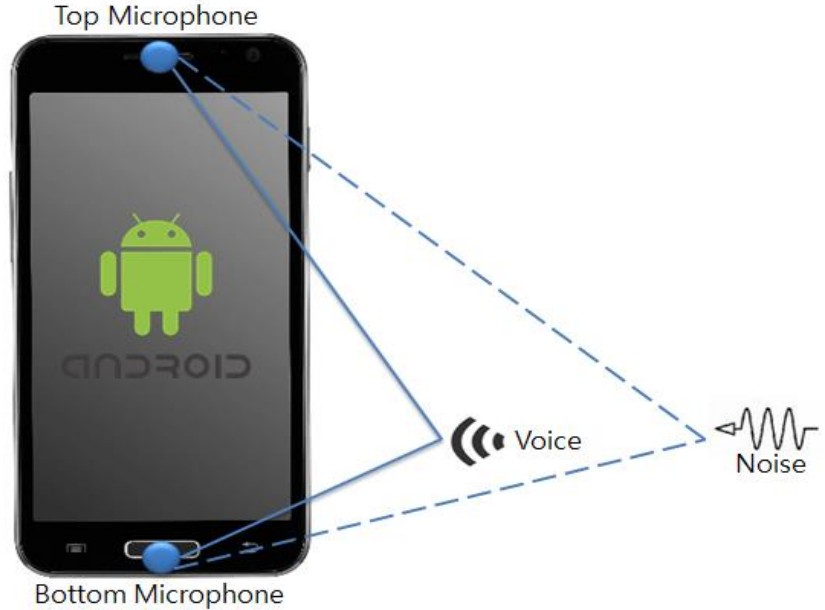

**Figure 1.** Measuring principle using microphone sensors.

Typically, three factors (i.e., sensitivity, frequency range, and signal conditioning) should be considered when selecting microphones for noise measurement. The sensitivity determines the voltage generated from each sound pressure level, and it is indicated in unit of mV/Pascal. For instance, the sound pressure of 1 Pascal Root Mean Square (RMS) is equivalent to the sound pressure of 94 dB, whereas 20 uPa is equivalent to 0 dB, which is generally considered to be the lowest listening level. Normally, a microphone with higher sensitivity is required when dealing with low-level noises. As for the frequency ranges, most of the microphones have the range of 20 Hz–20 KHz. Microphones with a better low-frequency response will have a larger size and cost. Lastly, signal conditioning converts the data collection device into an overall data collection system by directly linking various types of sensors (thermocouple, etc.) and signals (high-voltage signals, etc.) together. Both performance level and accuracy of data collection are influenced by the conditioning technique used.

In this study, the microphone sensors that can be used for the smart phones have been used to measure the noises and estimate directions. The noise level can be determined through three steps wherein the sounds will be entered via smart phone microphone sensors and then converted into decibels.

The noise level can be calculated with the decibel value obtained in Table 1. The code used in the algorithm is introduced in the Appendices A–D.

**Table 1.** Noise level measuring Algorithm.

| | |
|---|---|
| Step 1 | Read the noise with the embedded microphone sensor (getNoiseLevel of NoiseRecoder). It is then converted into electrical signal.<br>AudioRecord recorder = new AudioRecord (MediaRecorder. AudioSource. MIC, 44,100,<br>　　AudioFormat. CHANNEL_IN_DEFAULT,<br>　　AudioFormat. ENCODING_PCM_16BIT,<br>　bufferSize) |
| Step 2 | The noise of the electrical signal is removed with an EMA (Exponential Moving Average) filter, and then converted into amplitude (getAmplitude of NoiseDecibel).<br><br>$$EMA = EMA\_FILTER \times amp + (1.0\text{-}EMA\_FILTER) \times EMA$$ |
| Step 3 | Determine the decibel with the calculated amplitude via the decibel transformation formula.<br><br>$$dB = (20 \times Math.log10\ (pressure/REFERENCE)$$ |

## 2.2. Research Trend for Environmental Control Systems

At the Apple WorldWide Developers Conference (WWDC) 2014, 'Homekit' which allowed to control Apple household products through Siri (Speech Interpretation and Recognition Interface) was introduced by Apple themselves and its upgrade version which enabled the same control with Apple Watch was presented in the same year, automating home appliances control according to the life pattern of the user [4]. Similar technology was developed by LG electronics Korea who attempted to use a smartphone messenger for the control of their home appliances including washing machines, air conditioners, and audio/video systems [9–11].

The Radio Frequency Identification (RFID) technology is being used widely overseas for the environmental control systems using external sensors of voice recognition to control signals. Carol Rus, et al. [12] presented a voice-controlled smart house whereas Corcoran, et al. [13] introduced the Universal Plug-n-Play (UPnP) to allow wireless network users to experience various types of services with their PDAs or mobile devices. This technology enables an instant voice command to be transmitted to the home server user interface so that the user is free from the local imitations without relying on pre-arranged voice control. Meanwhile, Hwang, et al. [10] developed an RFID-based multi-user access control algorithm for the smart homes using UPnP. This algorithm monitors the access/activity of the users through the RFID tags worn by them. One of the major disadvantages of such a system was that a large number of RFID detectors had to be installed around the house. Helal, et al. [8] & Liau, et al. [9] proposed a wireless smart floor technology which could determine the position of residents with the pressure sensors embedded in the floor panels. Recently, Power Line Communication (PLC)-based smart environmental control systems are starting to receive attention [14–17].

## 2.3. Noise Measurement

The noise restriction level is 58dB for the apartments (2204) or 50dB for the multi-unit houses (2005) [18]. However, these standards were not as realistic as the statistics pertaining to the number of noise-related disputes had shown. The noise-related issues should be studied by considering the noise-measuring methods first. Usually, the noise levels are defined by the sound pressure levels or sound (acoustic) power levels set by the KS Measurement Standards (Table 2) [19].

**Table 2.** Noise measurements and standardization.

| KS Standards | KS Standard Name |
| --- | --- |
| KS A 300 | Sound term |
| KS A ISO 1683 | Recommended standard of sound—sound level |
| KS A ISO 1996-1-3 | Representation, measurement, and assessment methods of environmental noises stage 1: basic volumes, assessment process, etc. |
| KS A ISO 3740-7 | Sound<br>- Measuring method for the sound intensity of a power source<br>- Guidelines for using the standards |
| KS A ISO 7779 | Sound<br>- Measuring the air-borne spread noise emitted from the IT equipment |
| KS C IEC 60942 | Electro-acoustic—acoustic calibrator |
| KS C IEC 61672-1-2 | Electro-acoustic<br>- Sound level meter (noisemeter)<br>- Stage 1: standards |

*2.4. Direction Estimation Algorithm*

The speed of sound (sound velocity) in the air is approximately 340 m/s so that the time required for it to move 1 cm is approximately 29 us. However, as the minimum unit of measure that can detect the sound volume changes is 'ms', the sound waves cannot be measured. Thus, an algorithm which estimates the direction of sound based on vector computation was used.

The computation method of the algorithm using Table 3 is as follows: Calculate individual average values from three sound measurements taken and align them in descending order to perform a vector computation using the highest and second highest values. This allows estimation of the latitude and longitude of a noise-generating location along with its direction based on the same of the current smartphone's location. Meanwhile, Figure 2 describes a method of estimating the point of origin of the sound occurrence based on vector computation.

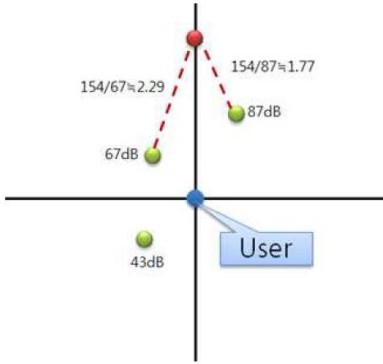

**Figure 2.** An example of direction estimation based on vector computation.

In this situation, the position of 'User' was set as 0 on the distance measuring plane and then the proportion value was calculated by using the highest and next highest noise values obtained from the two different locations [20,21]. Each proportion value can be calculated by adding those two noise values and then dividing it with the respective noise values separately. Finally, by using the proportion values of the two locations, the point of origins of the noise in question is estimated to define it as a target location for investigation.

The distance between the current position and the target location and its angle are calculated by converting them into radian values. Since one latitude and longitude degree covers approximately 111 km, their 7th (approximately 10 cm unit) and 8th (approximately 1 cm unit) values should be

compared to perform an accurate calculation of an indoor position. The direction estimating algorithm using the radian values is shown in Table 3.

**Table 3.** The direction estimating algorithm.

| Step 1 | Store the current position's latitude and longitude when measuring the noise. latitude = gps.getLatitude () longitude = gps.getLongitude () |
|---|---|
| Step 2 | In getAveValue of 'Comparison', align individual decibel values in descending order and perform vector computation by using the highest and second highest values to estimate the point of origin of the noise. Estimated point of origin of Noise = (Direction angle of the highest noise) ± ((Direction angle of the highest noise–Direction angle of the second-highest noise) × Proportion of two direction angles) |
| Step 3 | Calculate the direction angles and distance between current position and estimated location in getDirection and getDistance. Distance = acos(sin(current position latitude) × sin(Estimated location Latitude) + cos(Current Position Latitude) × cos(Estimated Location Latitude) × cos(Current Position Longitude–Estimated Location Longitude)) Direction = acos(sin(Estimated Location Latitude) sin(CurrentPosition Latitude) × cos (Distance)) /(cos (Current Position Latitude) × sin(Distance)) × (180/PI) |
| Step 4 | Calculate direction angle and distance between the current position and estimated location. |

*2.5. Korean Regulations for the Limits and Criteria with Regard to Inter-floor Noises in Astagement and Multi-unit Houses*

Table 4 is showing the regulations on the inter-floor noises: the direct noise levels are estimated based on one-minute-long equivalent sound level (Leq) and maximum sound level (Lmax) and the airborne noise levels are estimated based on a five-minute-long equivalent sound level. Nevertheless, a noise level of 5dB should be added to these restriction levels for both apartments and multi-unit houses constructed before June 30, 2005. Also, the measuring method of inter-floor noises should comply with the Measuring Standards of Business Site Noises (Official Test Standards for Noises and Vibrations) promulgated by the Minister of Environment, under Item. 2, Sec. 1, Art. 6) of the Environmental Examination and Inspection Act, which requires the measurements to be taken at more than one place for the duration of at least one hour. Regarding the equivalent sound levels of both durations (one and five minutes), the largest value should be taken in each case (Note. 3) and the noise should be considered excessive when it exceeds the Lmax more than three times in an hour.

Considering these regulations, this study proposes a measuring system with which the users will be able to collect the evidence for the noise-related issues conveniently.

**Table 4.** Regulations on the limits and criteria with regard to inter-floor noises in apartment and multiplex houses: Inter-floor noises.

| Classification of Inter-Floor Noises | | Criteria (Unit: dB(A)) | |
|---|---|---|---|
| | | Daytime (06:00–22:00) | Nighttime (22:00–06:00) |
| 1. Direct impact noises (Section 1, Article 2) | Equivalent Sound Level for 1 min (Leq) | 43 | 38 |
| | Max. Sound Level (Lmax) | 57 | 52 |
| 2. Airborne Noises (Section 2, Article 2) | Equivalent Sound Level for 5 min (Leq) | 45 | 40 |

*2.6. Comparison with Other System*

The accuracy of the apps used for noise measurement was assessed in Reference [22]. A smartphone (iPhone) was used and 37 apps from the 62 apps searched by using the keywords associated with 'noise measurement' were selected. From the measurement results, 31 apps exceeded the error range of ±2 dB, which is an error range of common noise meters, whereas 18 apps exceeded the error range of ±5 dB and the average error of 13 apps was ±11 dB. It was necessary to pay special attention to

most of these apps when using them as they did not seem to be completely fit for measuring noises. In Reference [23], the accuracy of the smartphone's noise measuring apps which can be used for the evaluation of noise generation in a working area was assessed to analyze their individual performance levels. As an experimental condition, the noise was gradually increased by 5 dB at a time within the range of 60–95 dB while using a pink noise having a frequency range of 20 kHz. The result obtained by using a total of 20 iPhone apps and Android apps showed that they had an average error of 0.07 dB when measuring noises within a particular range.

The existing study [22,23] had dealt with the accuracy of noise measuring apps commonly used for smartphones and found that many of them exceeded the error rate of actual noise meters or some of them produced higher accuracy only in a particular environment. Therefore, a step where the measurement values can be calibrated or compensated is required for improvements when measuring noises with a microphone sensor embedded in smartphones.

Meanwhile, research [24] used a noise measuring method using the difference in the loudness and proposed an algorithm which enables a robot to estimate the location of the target. The mobile robot was able to generate the coordinates of a sound source with its sensor and estimated the approximate bearing of the source by finding the direction while distinguishing four experimental areas distinguished by using a sound source tracking device.

Research [25] proposed a noise-tracking system using four microphones. By using a GCC (Generalized Cross Correlation)-PHAT (Phase Transform) method, the system calculates the difference in the sound arrival times between the microphones and the optimal location was determined by using Iterative Least Square based on the arrival time differences. The accuracy of the measurements was in centimeters after comparing the actual location and the estimated location.

Research [26] proposed a method of locating the positions of microphone and speaker and a talker in an ad hoc network. A performance evaluation was carried out by assuming a situation where a participant sets his/her laptop computer on a conference table to communicate with others through the built-in microphone set. In this case, the distance error was 22 cm between the actual and estimated positions of the microphone.

Research [27] proposed a method of determining the locations of microphone and sound source based on the difference in arrival times. The arrival times were calculated by arranging the flat-type microphone array in a grid form often referred to as a 'unit cube' to estimate locations.

Research [28] proposed an algorithm which allows a mobile robot to track the position of a sound source by using three microphones. The angles of the sound entering into each microphone were measured for the estimation purpose. This method exhibited 3–10% higher accuracy than the existing Threshold-based method. There were many instances of performing performance evaluations in a limited environment in the existing research [24–28] as it estimated the direction by using specialized equipment such as microphone array, etc. However, such a method has a problem when applying it to an actual household, so that another method which can be conveniently used at home with the common smartphones is often used. Thus, this study proposes a power line communication (PLC)-based solution where noises are measured through the microphone sensor replacing CCTV to estimate and record/save the direction of the sound source and sound level (dB) as evidence.

## 3. Digital Forensics System using PLC for Inter-Floor Noise Measurement: Focusing on PLC-based Android Solution Replacing CCTV Solution

The dispute over inter-floor noises in apartments or residential buildings has become a serious problem, often leading to a civil suit or even a criminal case. In most cases, disputing parties are civilians who attempt to collect evidence when arguing with their neighbors or settling a civil suit. It seems that there are not any effective solutions for this problem currently but some of the modern technologies may be useful in producing evidential materials such as noise measurements or a video including a scene causing a noise. This study focuses on digital forensics as a means of producing concrete evidence, explaining the causality of noise-occurring events, or analyzing the factors or elements

involved in the dispute. A number of IT or ICT technologies are being used for digital forensics along with other sophisticated analysis tools to guarantee the reliability of the collected evidence. However, due to the technological/technical complexity, digital forensic work is often performed by the firms specializing in it, which can be quite costly. This study aims to find an effective and efficient way of collecting evidence based on digital forensic technology and attempts to construct an analytical model with which the victims of noise can measure the noise level by themselves directly on the spot. The system design developed with Java/Android was made as simple and convenient as some of the recording/measuring devices available on the market, such as a vehicle black box, for example.

### 3.1. Structure of the Proposed System: Digital Forensics System Using PLC for Inter-Floor Noise Measurement

The system flow chart of the system developed with OPNET is shown in Figure 3: a single main PLC system is installed in each building whereas every household is equipped with the proposed system. The PLC system is connected to the lamp's power line along with the microphone sensor. To collect more accurate measurements and for easier management, the interior space should be partitioned and apply the same installation method to each space after assigning individual IDs to them [29–31]. Those Ids are shown in Table 5.

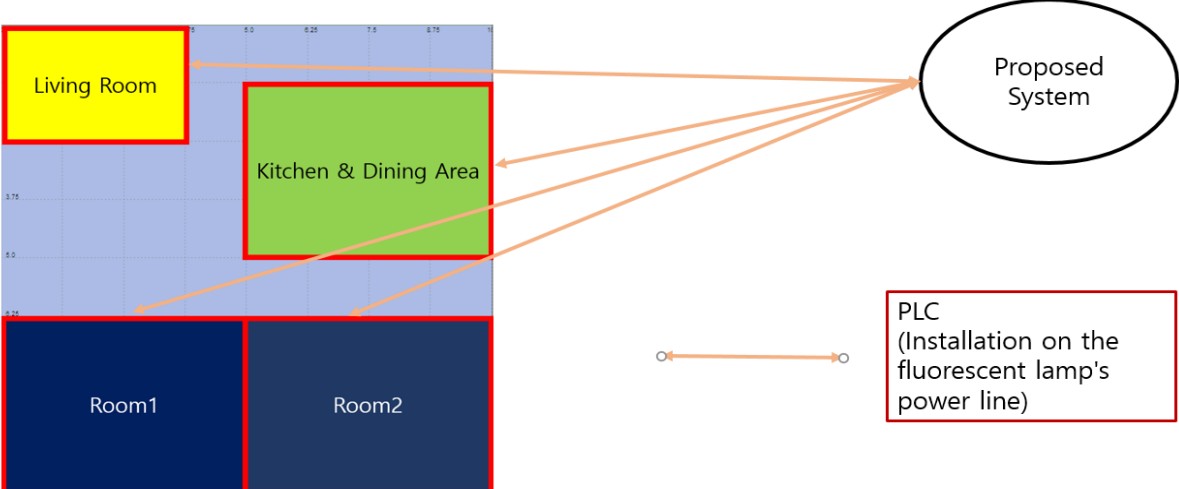

**Figure 3.** System flow map of the proposed system using OPNET Simulation.

**Table 5.** Management IDs.

| Area | | Management ID |
|---|---|---|
| | Living Room | 301 AREA 001 |
| | Kitchen and Dining Area | 301 AREA 002 |
| Apartment No. 301 | Room 1 | 301 AREA 003 |
| | Room 2 | 301 AREA 004 |
| | Room 3 | 301 AREA 005 |

The entire system is shown in Figure 4. Such a system is scheduled to be installed in each of the seven homes in a smart home-type four-story multi-unit villa for testing. The villa will be equipped with a solar ESS, home server, and other basic smart home systems adopting a smart grid technology.

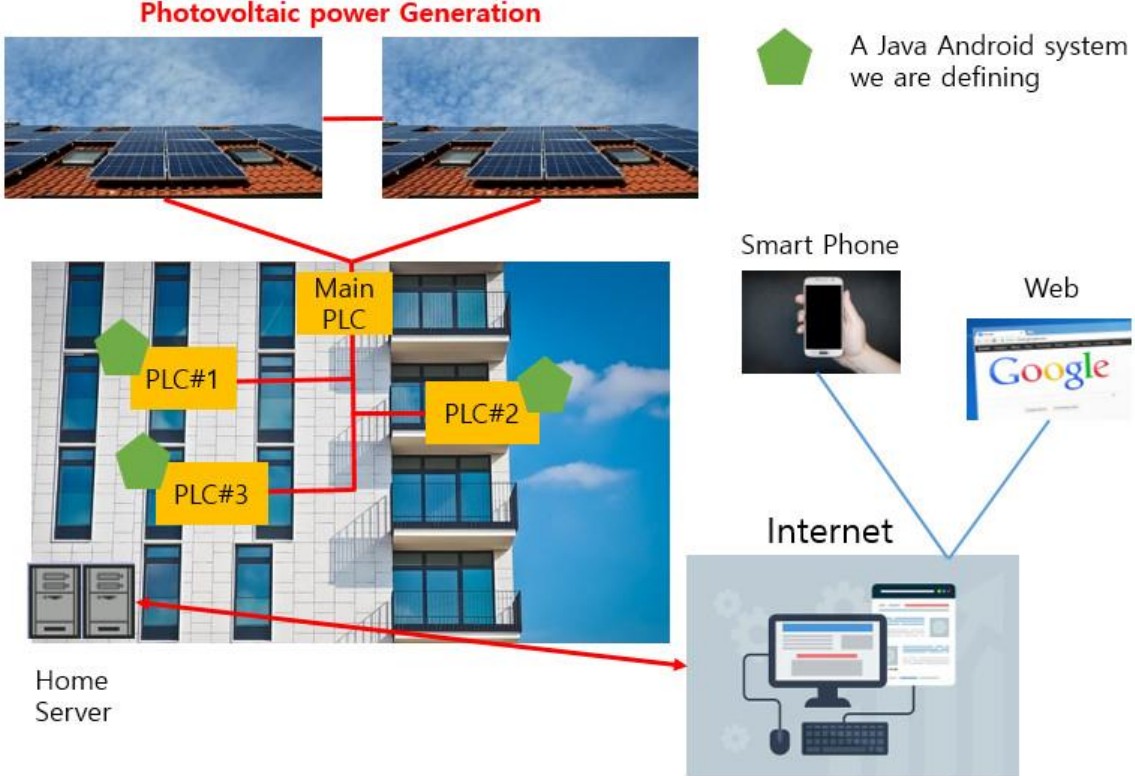

**Figure 4.** Aerial view of the system installation.

Figure 5 is describing the three individual stages of the proposed system: the physical stage includes microphone sensor and PLC system whose function is to collect the inter-floor noise data to maintain the home IoT system in an optimal state. The middle stage includes the sensor manager which controls the installed sensor and other relevant equipment mentioned in the first stage. The third stage contains the applications which support noise measuring and data storing functions. In this stage, texts and images are used to indicate the noise levels whereas a graph represents them as a recording service.

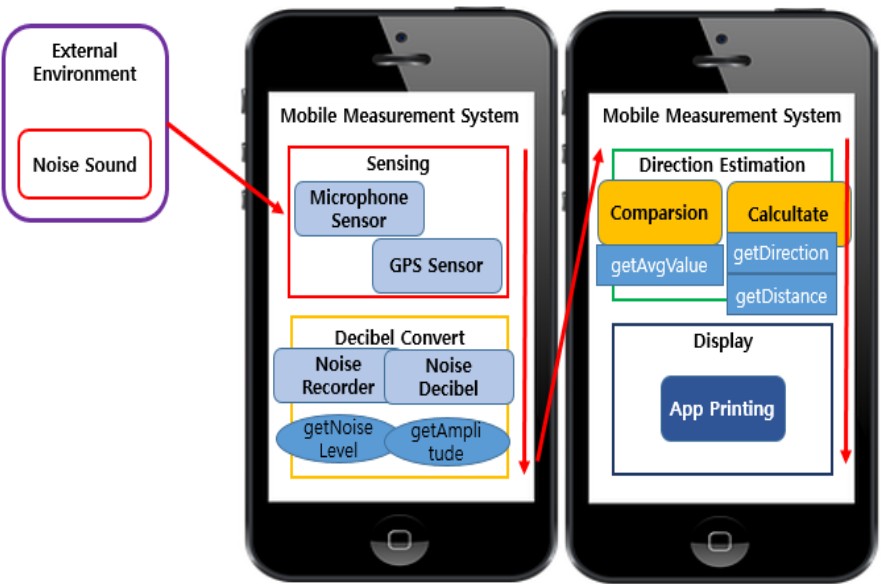

**Figure 5.** Structure of the proposed system.

### 3.2. The Algorithm Used for Measuring Inter-Floor Noises and Collecting Evidence: Application Stage

The algorithm specifically developed for the purpose of measuring inter-floor noises and collecting evidence is shown in Figure 6 where the reference value for the measurement has been set in dB first following government regulation. Then, the measurements taken at the site go through the filtering process to produce proposer noise data which will be subjected to a comparison with the reference value. The noise measuring process is repeated three times to guarantee reliability and if the measurement value is lower than the reference value, they will not be saved in the database. Such a process should be repeated three times or more.

```
PROCEDURE Noise Measurement
IF exist dB reference value then
        Read noise measuring

        IF enough noise measuring data then
                Filtering Stage function
                BEGIN
                        IF exceeded the reference value then
                                Saved noise data
                                IF repeat count = 3 then
                                        END
                                ELSE
                                        repeat count + 1
                                        Filtering step function
                                ENDIF
                        ELSE
                                do not saved noise data
                        ENDIF
                END
        ENDIF

ELSE
        Input the dB reference value
ENDIF
```

**Figure 6.** Algorithm for inter-floor noise measurement and evidence collection.

The filtering process is described in Figure 7 where the noise collected by the microphone sensor is converted into an electric signal and then into a dB unit indicating its loudness. In this case, the dB value is calculated by multiplying 20 by the common logarithm value calculated against the ratio betweeen reference loudness and measured loudness which are being respectively designated as A and B in the picture. The values change depending on the logarithmic function.

```
PROCEDURE Filtering Stage
Read the detect noise through microphone sensor

IF convert noise into electrical signals then
        Set converted signals calculated as dB through study algorithm
        IF the measurement is finished then
                save data
        ENDIF
ENDIF
```

**Figure 7.** Filtering process stage.

The noise measuring system model proposed in this study is shown Figure 8. The model was implemented with Java Android along with the UML. After initiating the system process, the time and date will be checked from Main Class. The reference value is set together this point and will be delivered to Input Class for storage, followed by noise measuring and recording through the microphone using the recorder function. The noise data being recorded is then passed to Noise Recorder Class amplifies the noise by using get Noise Level function and Filtering Class in Filtering Class receives the data and improves the quality of noise with its EMA filter (i.e., removing statics and amplifying the sound). Finally, Calculate Class generates the noise value in dB based on the 'converted amplitude value' presented by the calculator function, which indicates the loudness. This process is repeated three times or more to increase the reliability of the calculation.

```
PROCEDURE Main
Read textbox data(year,month,day,hour,minute,referenceValue)
BEGIN
        Start Function()
END

Start Function(referenceValue)
BEGIN
        NoiseRecoder Function(referenceValue,recodingData)
END

NoiseRecoder Function(referenceValue,recodingData)
BEGIN
        Check referenceValue
        Convert noise values to electrical signals
        Filtering Function(electricalSignals)
END

Filtering Function(electricalSignals)
BEGIN
        Set EMA Filter data
        Noise Value Filtering using EMA Filter
        Calculate Function(Filtering Noise data)
END

Calculate Function(Filtering Noise data)
BEGIN
        Determine if the conversion value is higher than the reference value
        Calculate the average of three measurements
        Output function(calculate data)
END

Output function(calculate data)
BEGIN
        printScreen
        End
END
```

**Figure 8.** Unified modeling language (UML) implemented with Java Android.

## 4. Performance Evaluation

It runs on a smartphone to set the reference value based on user input (an integer value) and allow the time and date data to be checked automatically. After the user confirms his/her reference value with the Toast function, he/she will be able to initiate the noise measuring process by clicking 'Start Measuring' box.

The screen on the left is showing the noises which did not exceed the reference value, indicating them with green bars whereas the screen on the right side is showing the noises which had exceeded the reference value, representing them with red bars. These results are then transmitted to the SD card for storage. The user can instantly and continuously check all the evaluation results just by looking into the contents saved the card.

To evaluate the noise measurement performance of the proposed system in this study, an experiment was conducted for the sound sources identical to actual noises.

### 4.1. Noise Measurement Experiment and Analysis

A total of 35 free test sound sources consisting of 22 domestic noises (9 types) and 13 other noises (7 types) were used as sound sources. These are distributed at MC2Method [32]. The types of sound sources are shown in Table 6. For the experiment, these sound sources were played by the computer in the lab and measured under the condition wherein the speaker, three Smart Phones, and the noise measuring apparatus had been equally spaced 50 cm apart. The noise measuring apparatus [33] used for the experiment was one that conformed to the Korean Industrial Standard (KS Standard), whereas the apps that had received more than 4.5 out of 5 in assessment points at the Android Google Market were selected as the noise measuring system [34] and sound level meter/noise detector [35].

**Table 6.** Types of test sound sources.

| Number | Types |
| --- | --- |
| 1 | Blender 1 |
| 2 | Blender 2 |
| 3 | Dryer 1 |
| 4 | Dryer 2 |
| 5 | Dryer 3 |
| 6 | Fan 1 |
| 7 | Fan 2 |
| 8 | Fan 3 |
| 9 | Fan 4 |
| 10 | Fan 5 |
| 11 | Heater 1 |
| 12 | Heater 2 |
| 13 | Heater 3 |
| 14 | Motor |
| 15 | Refrigerator |
| 16 | Shower |
| 17 | Stream 1 |
| 18 | Stream 2 |
| 19 | Vacuum |
| 20 | Water |
| 21 | Water Boiling 1 |
| 22 | Water Boiling 2 |
| 23 | Pink Noise |
| 24 | Train 1 |
| 25 | Train 2 |
| 26 | Under Water 1 |
| 27 | Under Water 2 |
| 28 | Under Water 3 |
| 29 | Waterfall |
| 30 | Waves 1 |
| 31 | Waves 2 |
| 32 | White Noise 1 |
| 33 | White Noise 2 |
| 34 | White Noise 3 |
| 35 | White Noise 4 |

Assuming that the measurement value of the noise measuring apparatus [33] had 100% accuracy (valuation criteria), the size of error between the criteria and each app was compared. It is possible to consider the accuracy to be high if the resulting measurements are identical.

Figures 9–11 show the results obtained from the measuring method proposed in this study using noise measuring apparatus [33], noise measuring apps [34], and db sound level meter/noise detectors [35] to evaluate the measuring performances.

Figure 9 shows the graph that displays the noise measurement results obtained from the proposed system together with the measurement values presented in Reference [33]. After measuring 35 test sound sources, all of the measurements except six were identical to the criteria, suggesting that the proposed system had an accuracy of about 82.8%. Moreover, the same system recorded a margin of error of less than ±3 dB, which is the general margin of error of a common noise measuring apparatus.

Figure 10 shows the noise measurement results from References [33,34]. At least 5 out of 35 measurements were identical to the criteria, thus having an accuracy of about 14.2%. Especially, Reference [34] exhibited a margin of error exceeding the normal range of ±5 dB of a common noise measuring apparatus. Figure 11 shows the measurements from References [33,35]. After measuring 35 test sound sources, there were no measurement values equivalent to the criteria (0% accuracy). There were some difficulties in measuring the actual noises as Reference [35] presented some irregular deviations compared to Reference [33]. Thus, the proposed system had the highest accuracy with approximately 83%.

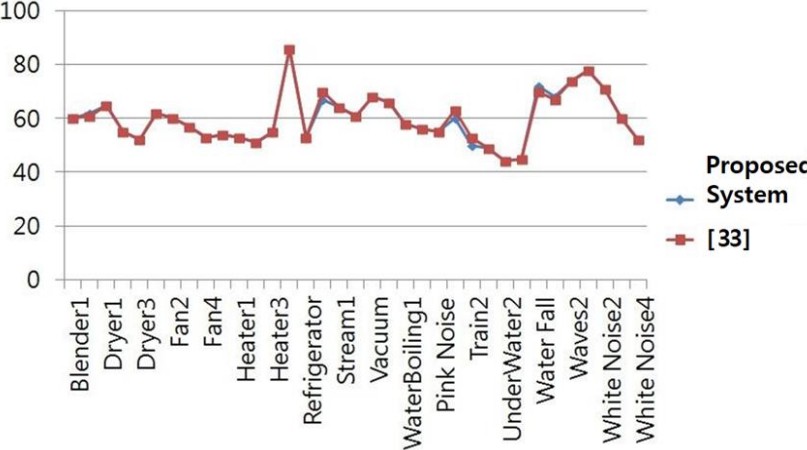

**Figure 9.** Measurements by the proposed system and from reference [33].

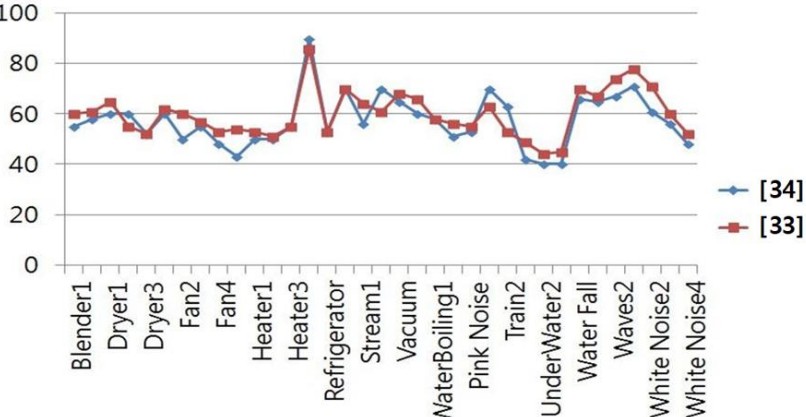

**Figure 10.** Measurements from references [33,34].

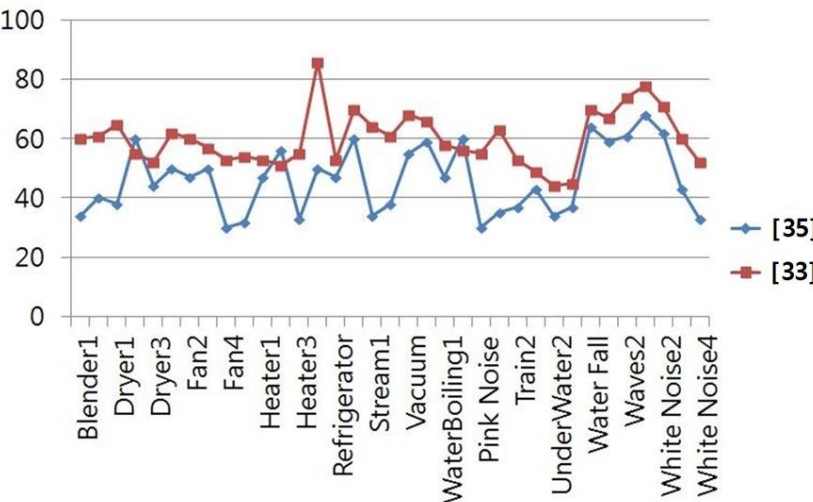

**Figure 11.** Measurements from references [33,35].

*4.2. Direction Estimation Experiment and Analysis*

To evaluate the performance of direction estimation, diameters of 1–5 m in the lab were first set for the experiment. Next, a direction that was to be the base direction and ten other directions with different angles were selected to find the original direction of the sound source by using the proposed system. Table 7 shows the results (accuracy) from this experiment.

**Table 7.** Result of direction estimation (accuracy).

| Number | Types | 1 m | 3 m | 5 m |
|---|---|---|---|---|
| 1 | North/5° | 355° | 4° | 9° |
| 2 | Northeast/47° | 62° | 40° | 32° |
| 3 | East/101° | 89° | 112° | 110° |
| 4 | Southeast/132° | 148° | 130° | 134° |
| 5 | Southeast/150° | 141° | 154° | 145° |
| 6 | South/200° | 210° | 198° | 193° |
| 7 | Southwest/236° | 223° | 241° | 227° |
| 8 | West/278° | 291° | 280° | 283° |
| 9 | Northwest/303° | 292° | 308° | 311° |
| 10 | Northwest/320° | 307° | 317° | 330° |
| Average Absolute Error | | 12.2° | 4.2° | 7.4° |

The results showed that the mean value of the absolute error was the smallest (4.2°) when the distance between the user and the noise was 3 m, compared to 7.4° at 5 m and 12.2° at 1 m. The large mean error and the resulting low accuracy at 1 m were deemed attributable to the too-close measuring distance.

Reference [30] explained that there was an error of 2–3 degrees when estimating the angles by using a microphone array and a mobile robot, and this research confirmed that the proposed method had the same performance level of direction estimation on the Android platforms.

The performance evaluation was carried out ahead of practical application, and the results revealed that the system had operated in a flexible way under the experimental environment. When comparing with the existing commercial noise measuring apps that provide simple real-time measurement or single measurement, the proposed system performs the measurement at least three times to increase the accuracy and has a function of estimating the origin of the noise based on them so that more accurate information can be provided to the user.

## 5. Conclusions and Future Work

Finding a proper and effective solution for the noise-related disputes between the residents living in an apartment or multi-unit house has been a difficult task for the government or judicial officers as there was not any reliable noise-measuring method available. Specifically, the pieces of equipment for collecting evidence (noises in this case) are too expensive, and the clients have to vacate their homes while measurements are taken.

While so many elements are complexly involved in the problems related to inter-floor noises, it is very difficult to conduct direct on-site analysis or construct analytical models that many rely on the paid services provided by private companies or organizations. Thus, we proposed a system that the clients can use for conveniently measuring inter-floor noises by installing the PLC system on the power line of a household fluorescent lamp. The PLC technology used in this study was described in detail in a previous study [15–17].

The proposed system is easy to handle as the language and modeling standards used for implementing the system are universal (Java Android & UML). Commercialization of this system is expected following the completion of patent registration. Meanwhile, the Digital Forensics System device can interact with users via diverse IoT sensors. Thus, the system that this paper deals with is set up in such a way that floor noise information is intelligently collected and utilized as evidence in the event of a serious issue in order to enable more effective interaction with the user. The system blocks power when the noise level does not meet the criteria but allows power when the noise reaches a certain level. Perhaps if a house is not equipped with Internet access, its lighting system will have some kind of "miniature" online access system.

The proposed system is scalable, and we were able to confirm that it works flexibly in the experimental environment. It provides application services through the server and the Database connected in the PLC network where a multiple number of microphone sensors collect noise data for later processing, which can be valuable in noise-related civil complaints. This system design is expected to be used widely in the field of digital forensics for noise-oriented conflicts or crimes as simply as the vehicle-mounted black boxes.

Compared with the existing commercial noise measuring apps that provide simple real-time measurement or single measurement, the proposed system performs the measurement at least 3 times to increase the accuracy and has a function of estimating the origin of the noise based on them so that more accurate information can be provided to the user.

The proposed system also exhibited accuracy of approximately 83% under a testing environment wherein other noise-measuring apps were involved. This was the highest level of accuracy compared with References [34,35]. The result of the direction estimation experiment has shown a mean margin of error of 4.2° at a distance of 3 m from the origin of the noise, confirming that the proposed system will be able to provide relatively accurate direction estimation performance on the mobile platforms and suggesting its usefulness under the household environment wherein the height of an individual home is about 2.5 m on average.

## 6. Discussion

Since 2005, the analysis methods for the multimedia voice evidence, especially the file-type voice evidence expressed with the binary system, have been quite limited and not systematized so that a systematized evidence evaluation method is required for a quick response. Although the human voice is often used in crimes using phones due to its anonymity, there is an identifier called 'voiceprint' which can distinguish/identify people and it is being used widely for the criminal cases. The voiceprint is also considered as a meaningful identifier in the field of biometrics along with fingerprint and DNA [36–39].

Thus, this study presented a digital forensics system which allows for collecting evidence to be used at the Korean court, which should be legal and free from forgery to guarantee the integrity of data. The system automatically records the noises exceeding the noise threshold (criterion) set in advance and the recorded data cannot be forged or falsified. The technological details of this part have been

excluded unavoidably as they are being included in a patent pending approval. There have been many neighborhood disputes over inter-floor noises in our country in almost every apartment or multi-unit house leading to a serious confrontation or even lethal retaliation involving murder. For this reason, the police are researching an efficient and legal method which allows them to collect evidence at a residence to which one or more noise-related complaints have been reported. The goal and motivation of the study are to monitor and collect legal evidence for a very malicious and persistent act of noise generation rather than a simple form of inter-floor noise often generated by the children's activities.

Currently, the noise criterion which can be a basis for claiming compensation is an average of 40 dB for a minute's measurement during the daytime and an average of 35 dB for the same period during the nighttime. It is necessary for the victim to send and keep the contents-certified letter asking the person generating the noises to refrain from making them. Additionally, the medical certificate issued by a psychiatrist is mostly used as direct evidence of damage.

**Author Contributions:** Conceptualization, M.-J.C.; Data curation, M.-J.C. and J.-H.H.; Formal analysis, M.-J.C.; Funding acquisition, J.-H.H.; Investigation, M.-J.C.; Methodology, M.-J.C. and J.-H.H.; Project administration, J.-H.H.; Resources, J.-H.H.; Software, M.-J.C. and J.-H.H.; Supervision, J.-H.H.; Visualization, J.-H.H.; Writing–original draft, M.-J.C. and J.-H.H.; Writing–review and editing, J.-H.H.

**Funding:** This work was supported by the National Research Foundation of Korea (NRF) grant funded by the Korea government (MSIT) (No.2017R1C1B5077157). Also, this research was supported by the MSIT (Ministry of Science and ICT), Korea, under the ITRC (Information Technology Research Center) support program (IITP-2019-2014-1-00743) supervised by the IITP (Institute for Information & communications Technology Planning and Evaluation).

**Acknowledgments:** The first part of this paper [21] was presented in an Oral Session in 2017 2nd Special Issue International Workshop at Namseoul University, Mar. 23–24 (2017). I am grateful to two anonymous commentators who have contributed to the enhancement of the paper's completeness with their valuable suggestions at the Workshop.

**Conflicts of Interest:** The authors declare no conflict of interest.

## Appendix A. Precedent in the ROK

One of the notable precedents concerning the dispute over the inter-floor problem can be traced back to a case in 2013. In this case, the downstairs residents were suffering from the noise (maximum 72.8 dB) arising from the dumbbells used by the upstairs residents and eventually partially won the suit in February 2014. The court ordered indemnification of 500,000 Korean won to the upstairs residents as they ruled that the noises had exceeded the bearable level. The winning party submitted the noise measurements taken with the noise-measuring application embedded in a smartphone along with the videos recording the deliberate acts of causing noises by bouncing a basketball or rolling dumbbells at night. However, the losing party raised a question about these videos and filed a lawsuit for the damage caused by invasion of privacy, which lead to the counteraction by the downstairs residents who claimed that the noises were continuing.

The 16th Civil Case Court (Daejeong Dist) has recently made a final judgment on this matter by taking both sides: ordering the downstairs residents to pay 500,000 Korean won as a compensation for the infringement of privacy to each of a couple upstairs, who in turn was ordered to pay 2 million Korean won to each of a couple downstairs for not stopping the noises. The court's ruling was based on their decision on the videos in question. First, the court agreed that in such a civil suit, the purpose of finding truth does not take precedence over protection of private lives so that the activities of upstairs residents should be kept in secret as their right. The court also pointed out that the defendant did not just submit the video evidence to the court but also allowed it to be broadcasted on the news, which was a violation of privacy. Nevertheless, the court limited the compensation under 500,000 won considering the fact that it was quite difficult for the defendant to collect the evidence of intentional inter-floor noise generation and they, in fact, tried to solve the dispute by reporting the problem to the apartment management office or police [40].

On the other hand, the counteraction taken by the downstairs residents concerning the everlasting noises was accepted as well. The court admitted that although the measurements taken with a mobile phone could be different, the couple downstairs had measured the noise levels for a long period of time with it and the results did not significantly deviate from the ones that had been taken with the specialized noise-measuring equipment used by the expert, adding the fact that the plaintiff had partially lost their argument in the previous court but did not make any effort to reduce noise even after receiving an injunction ordering them to stop the noises. The 2 million Korean won compensation was set based on these facts but in the end the downstairs residents moved to another home prior to the final date of argument, dissolving their neighborhood relationship.

**Appendix B. Inter-Floor issue in New York, USA**

The inter-floor noise-related problems are also serious in New York where most of the apartment floors are made of wood and many buildings have been constructed with bricks instead of concrete, allowing the noises from upstairs to be transmitted to the downstairs without much reduction. Another major cause of such noise generation is New Yorkers' party culture often involving continuous stomping or fussing throughout the night. What do the neighbors usually do in this case?

Although they are known to be impatient people, it is not customary for them to go straight to upstairs but instead, they make a complaint to the building supervisor or the landlord by sending a letter, it is better to keep such a record as it could become evidence when things get ugly. Upon receipt of the complaint, they give a warning to the noise-causing resident first and then report to the police if the situation is not improved. If the police are dispatched, he/she can be often be arrested or fined if there was proof of a complaint letter [41].

However, if such a case persists, people usually choose to move out as New Yorkers often do so, unlike Koreans. They do make a complaint to a real estate agent but move out later after being unable to cope with the noise. There is a contract term 'quiet enjoyment' when signing a rental agreement and this is considered as an implicit agreement that the signing party has the right to complain about the noises. It is also normal that the resident upstairs moves out after becoming an eyesore among the residents.

**Appendix C. Inter-Floor Issue in Tokyo, Japan**

It is estimated that about 40% of Japanese people are living in an apartment or a multi-unit house. Especially in a large city like Tokyo, most of them are living in an apartment building due to the very high housing cost. Nevertheless, different from the ROK, the inter-floor noise problem has not become a major social issue in their society yet as one of their national traits is to avoid causing any inconveniences to the others and live rather quietly while considering or respecting each other. Many Koreans who experienced their lives in an apartment house for the first time in Japan would agree to an observation, "Japanese maintain a surprisingly quiet lifestyle except when they are running for the toilet urgently." This may have been a strict education from their parents or elders starting at an early age. Their noise management in their homes is usually meticulous so that they watch TV with at the lowest volume or refrain from making a fuss at home. Some of the apartment complexes even restrict pets or lay carpets to reduce the noise generation. Despite such efforts, there still exist inter-floor noise-related disputes which can be found in the newspapers sometimes. The provision Article 1, Paragraph 14 states that those who cause inconvenience to the neighbors by making loud noises with their voices, musical instruments, or radios against the restraining order given by the public officials could be subjected to a fine or confinement but in most cases, this provision remains as a warning and other means of penalty will be given to correct their attitude among the residents themselves [42].

Thus, in such an environment, those who actually report inter-floor noise are scarce and those who really wish to avoid this kind of problem and have enough money prefer a single-family house where their privacy can be guaranteed. However, like many other countries, the majority of Japanese people

cannot enjoy such a benefit, so they live in harmony by considering their neighbors and maintaining good manners.

**Appendix D. Inter-Floor Issue in Sydney, Australia**

Having some regional differences, the Australian noise-related regulation is quite strict and covers apply to a wide range. For example, it is usually mentioned in almost every housing lease and the residents are required to fully understand it before signing the lease. Those who do not keep to it will have to accept the penalties. The regulation clearly states the daily time range and the types of noises allowed: 7 am to 7 pm for weekdays or 9 am to 7 pm for the weekend/holidays for ordinary noises. On the other hand, the party noises are acceptable until midnight on Fridays or Saturdays [43].

The residents do not confront directly where there is a dispute over noises between them but instead, they write a complaint letter to the opponent first. If this does not work, they ask the community advisory committee or housing management office to send an official warning letter. As a final measure to resolve the matter, they take legal action or call the police who are allowed to exercise physical force or issue a fine of two to four hundred Australian dollars (Approximately 250,000 Korean won) [44–47]. A more serious case can lead to a legal battle. However, there is a group called 'Neighbor Watch', who is scarier than the police. They refer to the Australians who report the neighbor-bothering acts or crimes like a noisy party for the residents even if there had not been any direct damages to them. Although Australian life consisting of emigrants may seem quite free and relaxing, the citizens are actually sticking to the rules and boundaries set by themselves.

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
