# Peer review of "Digital Forensics System Using PLC for Inter-Floor Noise Measurement: Detailing PLC-Based Android Solution Replacing CCTV-based Solution"

_electronics, doi:10.3390/electronics8101091_

Round 1

Reviewer 1 Report

The paper deals with an application regarding the measurement and localization of inter-floor noises (generated by appliances) in residential environments. This paper is actually a description of an application carried out by the Authors, rather than a research approach to a problem. This make the whole work weak under the research aspects. 

Author Response

Reply -

I am grateful for your comment. I’ve tried to answer all of your comments from the readers’ perspective with the help of a native English speaker. The additions and changes made have been highlighted in blue. I’d like to respectfully request your re-review if possible.

Does the introduction provide sufficient background and include all relevant references?

( ) (x) ( ) ( )

Is the research design appropriate?

( ) ( ) ( ) (x)

Are the methods adequately described?

(x) ( ) ( ) ( )

Are the results clearly presented?

( ) ( ) (x) ( )

Are the conclusions supported by the results?

(x) ( ) ( ) ( )

Comments and Suggestions for Authors

The paper deals with an application regarding the measurement and localization of inter-floor noises (generated by appliances) in residential environments. This paper is actually a description of an application carried out by the Authors, rather than a research approach to a problem. This make the whole work weak under the research aspects.

Reply -

Thank you for your detailed review and appropriate comment. I’ve made the research design clearer for your review. The contribution and the significance of this research work or the application were to be mentioned in the Special Issue (Intelligent Closed-Circuit Television and Applications) but they have been supplemented. At the same time, many parts of the manuscript have been re-written or revised to increase clarity by explaining some of the related works and detailing the research aspects while discussing with a native English proofreader. All the changes additions made have been highlighted in blue. Hence, I respectfully request your re-review if possible.

Add) In the Republic of Korea, the number of inter-floor noise-related civil complaints received by the Neighborhood Inter-Floor Noise Complaint Center affiliated with the Ministry of Environment has increased from 8,795 in 2012 to 28, 231 in 2018. Although this organization is actively stepping in to mediate the dispute between neighbors, they do not seem to have any clear means of solving the problem as the causality involved in the noise-related issues are mostly due to the structural problems. The psychological damage can be compensated through a civil suit finding that the noises had exceeded the usual level one can tolerate but such a process requires a long judicial procedure.

Digital forensics is a new area of security service where the relevant fact of a certain activity can be invested gated and proved based on digital data by using information equipment as a medium. It is being used for criminal investigations by the national investigation authorities such as police, prosecution, etc. and its necessity is increasingly recognized in the private sectors including financial or security companies as well. For example, this technic is quite useful in collecting legal evidence, preventing internal information or strengthening internal security of auditing. Meanwhile, digital forensics is largely performed through the process of evidence collection, analysis, and submission. This study has focused on the evidence collection process.

Add) The dispute over inter-floor noises in apartments or residential buildings has become a serious problem often leading a civil suit or even a criminal case. In most cases, disputing parties are civilians who attempt to collect evidence when arguing with their neighbors or settling a civil suit. It seems that there are not any effective solutions for this problem currently but some of the modern technologies may be useful in producing evidential materials such as noise measurements or a video including a scene causing a noise. This study focuses on digital forensics as a means of producing concrete evidence, explaining the causality of noise-occurring events, or analyzing the factors or elements involved in the dispute. A number of IT or ICT technologies are being used for digital forensics along with other sophisticated analysis tools to guarantee the reliability of the collected evidence. However, due to the technological/technical complexity, digital forensic work is often performed by the firms specializing in it, which can be quite costly. This study aims to find an effective and efficient way of collecting evidence based on digital forensic technology and attempts to construct an analytical model with which the victims of noise can measure the noise level by themselves directly on the spot. The system design developed with Java/Android was made simple and convenient as some of the recording/measuring devices available on the market such a vehicle black box, for example.

Add) 6. Discussion

This study presented a digital forensics system which allows collecting evidence to be used at the Korean court, which should be legal and free from forgery to guarantee the integrity of data. The system automatically records the noises exceeding the noise threshold (criterion) set in advance and the recorded data cannot be forged or falsified. The technological details of this part have been excluded unavoidably as they are being included in a patent pending for approval. There have been many neighborhood disputes over inter-floor noises in our country in almost every apartment or multi-unit house leading to a serious confrontation or even lethal retaliation involving murder. For this reason, the police are researching an efficient and legal method which allows them t collect evidence at a residence to which one or more noise-related complaint has been reported. The goal and motivation of the study are to monitor and collect legal evidence for a very malicious and persistent act of noise generation rather than a simple form of inter-floor noise often generated by the children’s activities.

Currently, the noise criterion which can be a basis for claiming compensation is an average of 40dB for a minute’s measurement during the daytime and an average of 35dB for the same period during the night time. It is necessary for the victim to send and keep the contents-certified letter asking the person generating the noises to refrain from making them. Additionally, the medical certificate issued by a psychiatrist is mostly used as direct evidence of damage.

APPENDIX 1. A Precedent in the ROK

One of the notable precedents concerning the dispute over the inter-floor problem can be traced back to the case in 2013. In this case, the residents downstairs were suffering from the noise (max. 72.8dB) arising from the dumbbells used by the upstairs residents and partially won the suit in February 2014 eventually. The court ordered indemnification of 500,000won to the residents upstairs as they ruled that the noises had exceeded the bearable level. The winning party submitted the noise measurements taken with the noise-measuring application embedded in a smartphone along with the videos recording the deliberate acts of causing noises by bouncing a basketball or rolling dumbbells at night. However, the losing party raised a question about these videos and filed a lawsuit for the damage caused by invasion of privacy, which lead to the counteraction by the residents downstairs who claimed that the noises were continuing.

The 16th Civil Case Court (Daejeong Dist.) has recently made a final judgment on this matter by taking both sides: ordering the residents downstairs to pay 500,000won as a compensation for the infringement of privacy to each of a couple upstairs, who in turn was ordered to pay 2 million own to each of a couple downstairs for not stopping the noises. The court’s ruling was based on their decision on the videos in question. First, the court agreed that in such a civil suit, the purpose of finding truth does not take precedence over protection of private lives so that the activities of upstairs residents should be kept in secret as their right. The court also pointed out that the defendant did not just submit the video evidence to the court but also allowed it to be broadcasted on the news, which was a violation of privacy. Nevertheless, the court limited the compensation under 500,000 won considering the fact that it was quite difficult for the defendant to collect the evidence of intentional inter-floor noise generation and they, in fact, tried to solve the dispute by reporting the problem to the apartment management office or police [29].

On the other hand, the counteraction taken by the downstairs residents concerning the everlasting noises was accepted as well. The court admitted that although the measurements taken with a mobile phone could be different, the couple downstairs had measured the noise levels for a long period of time with it and the results did not deviate from the ones that had been taken with the specialized noise-measuring equipment used by the expert significantly, adding the fact that the plaintiff had partially lost their argument in the previous court but did not make any effort to reduce noise even after receiving an injunction ordering them to stop the noises. The 200 million won compensation was set based on these facts but in the end the residents downstairs moved to another home prior to the final date of argument, dissolving their neighborhood relationship.

APPENDIX 2. Inter-Floor issue in New York at USA

 The inter-floor noise-related problems are also serious in New York where most of the apartment floors are made of wood and many of buildings have been constructed with bricks instead of concrete, allowing the noises from upstairs to be transmitted to the downstairs without much reduction. Another major cause of such a noise generation is New Yorkers’ party culture often involving continuous stomping or fussing throughout the night. What do the neighbors usually do in this case?

Although they are known to be impatient people, it is not customary for them to go up straight to upstairs but instead, they make a complaint to the building supervisor or the landlord by sending a letter it is better to keep such a record as it could become evidence when the things get ugly. Upon receipt of the complaint, they give a warning to the noise-causing resident first and then report to the police if the situation is not improved. If the police are dispatched, he/she can be often arrested or fined if there was a proof of a complaint letter [30].

However, if such a case persists, people usually choose to move out as New Yorkers often do so unlike Koreans. They do make a complaint to a real estate agent but move out later after being unable to cope with the noise. There is a contract term ‘quiet enjoyment’ when signing a rental agreement and this is considered as an implicit agreement that the signing party has the right to complain about the noises. It is also normal that the resident upstairs moves out after becoming an eyesore among the residents.

APPENDIX 3. Inter-Floor Issue in Tokyo at Japan

It is being estimated that about 40% of Japanese are living in an apartment or a multi-unit house. Especially, in a large city like Tokyo, most of them are living in an apartment building due to the very high housing cost. Nevertheless, different from the ROK, the inter-floor noise problem has not become a major social issue in their society yet as one of their national traits is to avoid causing any inconveniences to the others and live rather quietly while considering or respecting each other. Many Koreans who experienced their lives in an apartment house for the first time in Japan would agree to an observation, “Japanese maintain a surprisingly quiet lifestyle except when they are running for the toilet urgently.” This may have been a strict education from their parents or elders starting at an early age. Their noise management in their homes is usually meticulous so that they watch TV with at the lowest volume or refrain from making a fuss at home. Some of the apartment complexes even restrict pets or lay carpets to reduce the noise generation. Despite such efforts, there still exist inter-floor noise-related disputes which can be found in the newspapers sometimes. The provision Article 1 Paragraph 14 states that those who cause inconvenience to the neighbors by making loud noises with their voices, musical instruments, or radios against the restraining order given by the public officials could be subjected to a fine or confinement but in most cases, this provision remains as a warning and other means of penalty will be given to correct their attitude among the residents themselves [31]. Thus, in such an environment, those who actually report inter-floor noise are scarce and those who really wish to avoid this kind of problem and have enough money prefer a single-family house where their privacy can be guaranteed. However, like many other countries, the majority of Japanese people cannot enjoy such a benefit so that they live in harmony by considering their neighbors and maintaining good manners.

APPENDIX 4. Inter-Floor Issue in Sydney at Australia

Having some regional differences, the Australian noise-related regulation is quite strict and covers apply to a wide range. For example, it is usually mentioned in almost every housing lease and the residents are required to fully understand it before signing the lease. Those who do not keep it will have to accept the penalties. The regulation clearly states the daily time range and the types of noises allowed: 7 am to 7 pm for week days or 9 am to 7 pm for the weekend/holidays for ordinary noises. On the other hand, the party noises are acceptable until midnight on Fridays or Saturdays [32].

   The residents do not confront directly where there is a dispute over noises between them but instead, they write a complaint letter to the opponent first. If this does not work, they ask the community advisory committee or housing management office to send an official warning letter. As a final measure to resolve the matter, they take legal action or call the police who are allowed to exercises physical force or issue a fine of two to four hundred Australian dollars (Approx. 250,000 Korean won). A more serious case can lead to a legal battle. However, there is a group called ‘Neighbor Watch’, who is scarier than the police. They refer to the Australians who report the neighbor-bothering acts or crimes like a noisy party for the residents even if there had not been any direct damages to them. Although Australian life consisting of emigrants may seem quite free and relaxing, the citizens are actually sticking to the rules and boundary set by themselves.

APPENDIX 5. Java Android Source Code: 1. Main Java Android Source Code

Reviewer 2 Report

The research is interesting for the measurement. One issue that might appear in court is that the complainant was tampering with the system. Perhaps one can add a sound source with noise to exceed the thresholds locally. It is not clear how this is solved.

Also there is a privacy issue since the contents of speech might also be recorded. This issue is not handled in the paper and should be discussed.

Author Response

Reply -

I appreciate your comment. This study presented a digital forensics system which allows collecting evidence to be used at the Korean court, which should be legal and free from forgery to guarantee the integrity of data. The system automatically records the noises exceeding the noise threshold (criterion) set in advance and the recorded data cannot be forged or falsified. The technological details of this part have been excluded unavoidably as they are being included in a patent pending for approval. There have been many neighborhood disputes over inter-floor noises in our country in almost every apartment or multi-unit house leading to a serious confrontation or even lethal retaliation involving murder. For this reason, the police are researching an efficient and legal method which allows them t collect evidence at a residence to which one or more noise-related complaint has been reported. The goal and motivation of the study are to monitor and collect legal evidence for a very malicious and persistent act of noise generation rather than a simple form of inter-floor noise often generated by the children’s activities.

  Currently, the noise criterion which can be a basis for claiming compensation is an average of 40dB for a minute’s measurement during the daytime and an average of 35dB for the same period during the night time. It is necessary for the victim to send and keep the contents-certified letter asking the person generating the noises to refrain from making them. Additionally, the medical certificate issued by a psychiatrist is mostly used as direct evidence of damage. The content of a precedent written and disclosed by the court for the research by the law school students or members of the judicial system is being introduced in the reference. This material was translated into English and attached.

Add)

APPENDIX 1. A Precedent in the ROK

One of the notable precedents concerning the dispute over the inter-floor problem can be traced back to the case in 2013. In this case, the residents downstairs were suffering from the noise (max. 72.8dB) arising from the dumbbells used by the upstairs residents and partially won the suit in February 2014 eventually. The court ordered indemnification of 500,000won to the residents upstairs as they ruled that the noises had exceeded the bearable level. The winning party submitted the noise measurements taken with the noise-measuring application embedded in a smartphone along with the videos recording the deliberate acts of causing noises by bouncing a basketball or rolling dumbbells at night. However, the losing party raised a question about these videos and filed a lawsuit for the damage caused by invasion of privacy, which lead to the counteraction by the residents downstairs who claimed that the noises were continuing.

The 16th Civil Case Court (Daejeong Dist.) has recently made a final judgment on this matter by taking both sides: ordering the residents downstairs to pay 500,000won as a compensation for the infringement of privacy to each of a couple upstairs, who in turn was ordered to pay 2 million own to each of a couple downstairs for not stopping the noises. The court’s ruling was based on their decision on the videos in question. First, the court agreed that in such a civil suit, the purpose of finding truth does not take precedence over protection of private lives so that the activities of upstairs residents should be kept in secret as their right. The court also pointed out that the defendant did not just submit the video evidence to the court but also allowed it to be broadcasted on the news, which was a violation of privacy. Nevertheless, the court limited the compensation under 500,000 won considering the fact that it was quite difficult for the defendant to collect the evidence of intentional inter-floor noise generation and they, in fact, tried to solve the dispute by reporting the problem to the apartment management office or police [29].

On the other hand, the counteraction taken by the downstairs residents concerning the everlasting noises was accepted as well. The court admitted that although the measurements taken with a mobile phone could be different, the couple downstairs had measured the noise levels for a long period of time with it and the results did not deviate from the ones that had been taken with the specialized noise-measuring equipment used by the expert significantly, adding the fact that the plaintiff had partially lost their argument in the previous court but did not make any effort to reduce noise even after receiving an injunction ordering them to stop the noises. The 200 million won compensation was set based on these facts but in the end the residents downstairs moved to another home prior to the final date of argument, dissolving their neighborhood relationship.

APPENDIX 2. Inter-Floor issue in New York at USA

 The inter-floor noise-related problems are also serious in New York where most of the apartment floors are made of wood and many of buildings have been constructed with bricks instead of concrete, allowing the noises from upstairs to be transmitted to the downstairs without much reduction. Another major cause of such a noise generation is New Yorkers’ party culture often involving continuous stomping or fussing throughout the night. What do the neighbors usually do in this case?

Although they are known to be impatient people, it is not customary for them to go up straight to upstairs but instead, they make a complaint to the building supervisor or the landlord by sending a letter it is better to keep such a record as it could become evidence when the things get ugly. Upon receipt of the complaint, they give a warning to the noise-causing resident first and then report to the police if the situation is not improved. If the police are dispatched, he/she can be often arrested or fined if there was a proof of a complaint letter [30].

However, if such a case persists, people usually choose to move out as New Yorkers often do so unlike Koreans. They do make a complaint to a real estate agent but move out later after being unable to cope with the noise. There is a contract term ‘quiet enjoyment’ when signing a rental agreement and this is considered as an implicit agreement that the signing party has the right to complain about the noises. It is also normal that the resident upstairs moves out after becoming an eyesore among the residents.

APPENDIX 3. Inter-Floor Issue in Tokyo at Japan

It is being estimated that about 40% of Japanese are living in an apartment or a multi-unit house. Especially, in a large city like Tokyo, most of them are living in an apartment building due to the very high housing cost. Nevertheless, different from the ROK, the inter-floor noise problem has not become a major social issue in their society yet as one of their national traits is to avoid causing any inconveniences to the others and live rather quietly while considering or respecting each other. Many Koreans who experienced their lives in an apartment house for the first time in Japan would agree to an observation, “Japanese maintain a surprisingly quiet lifestyle except when they are running for the toilet urgently.” This may have been a strict education from their parents or elders starting at an early age. Their noise management in their homes is usually meticulous so that they watch TV with at the lowest volume or refrain from making a fuss at home. Some of the apartment complexes even restrict pets or lay carpets to reduce the noise generation. Despite such efforts, there still exist inter-floor noise-related disputes which can be found in the newspapers sometimes. The provision Article 1 Paragraph 14 states that those who cause inconvenience to the neighbors by making loud noises with their voices, musical instruments, or radios against the restraining order given by the public officials could be subjected to a fine or confinement but in most cases, this provision remains as a warning and other means of penalty will be given to correct their attitude among the residents themselves [31]. Thus, in such an environment, those who actually report inter-floor noise are scarce and those who really wish to avoid this kind of problem and have enough money prefer a single-family house where their privacy can be guaranteed. However, like many other countries, the majority of Japanese people cannot enjoy such a benefit so that they live in harmony by considering their neighbors and maintaining good manners.

APPENDIX 4. Inter-Floor Issue in Sydney at Australia

Having some regional differences, the Australian noise-related regulation is quite strict and covers apply to a wide range. For example, it is usually mentioned in almost every housing lease and the residents are required to fully understand it before signing the lease. Those who do not keep it will have to accept the penalties. The regulation clearly states the daily time range and the types of noises allowed: 7 am to 7 pm for week days or 9 am to 7 pm for the weekend/holidays for ordinary noises. On the other hand, the party noises are acceptable until midnight on Fridays or Saturdays [32].

   The residents do not confront directly where there is a dispute over noises between them but instead, they write a complaint letter to the opponent first. If this does not work, they ask the community advisory committee or housing management office to send an official warning letter. As a final measure to resolve the matter, they take legal action or call the police who are allowed to exercises physical force or issue a fine of two to four hundred Australian dollars (Approx. 250,000 Korean won). A more serious case can lead to a legal battle. However, there is a group called ‘Neighbor Watch’, who is scarier than the police. They refer to the Australians who report the neighbor-bothering acts or crimes like a noisy party for the residents even if there had not been any direct damages to them. Although Australian life consisting of emigrants may seem quite free and relaxing, the citizens are actually sticking to the rules and boundary set by themselves.

APPENDIX 5. Java Android Source Code: 1. Main Java Android Source Code

Also there is a privacy issue since the contents of speech might also be recorded. This issue is not handled in the paper and should be discussed.

Reply -

I appreciate your comment.

Add) 6. Discussion

This study presented a digital forensics system which allows collecting evidence to be used at the Korean court, which should be legal and free from forgery to guarantee the integrity of data. The system automatically records the noises exceeding the noise threshold (criterion) set in advance and the recorded data cannot be forged or falsified. The technological details of this part have been excluded unavoidably as they are being included in a patent pending for approval. There have been many neighborhood disputes over inter-floor noises in our country in almost every apartment or multi-unit house leading to a serious confrontation or even lethal retaliation involving murder. For this reason, the police are researching an efficient and legal method which allows them t collect evidence at a residence to which one or more noise-related complaint has been reported. The goal and motivation of the study are to monitor and collect legal evidence for a very malicious and persistent act of noise generation rather than a simple form of inter-floor noise often generated by the children’s activities.

Currently, the noise criterion which can be a basis for claiming compensation is an average of 40dB for a minute’s measurement during the daytime and an average of 35dB for the same period during the night time. It is necessary for the victim to send and keep the contents-certified letter asking the person generating the noises to refrain from making them. Additionally, the medical certificate issued by a psychiatrist is mostly used as direct evidence of damage.

Round 2

Reviewer 1 Report

I would like to thank the Authors to have revised the paper. Unfortunately, the paper is still weak under the scientific point of view, but good in terms of the application presentation. In fact, many words have been spent for describing the problem, but very few for the implemented technical solutions. Moreover, nothing is written about the microphone equalization, neither about the microphones electrical characteristics. 
It is also not clear how the receiver converts the sounds into the dB(A) values, mentioned in table 3. 
The direction estimation algorithm is not mentioned at all, but only showing the results. This makes the description weak under the scientific point of view.

Author Response

 (x)  Moderate English changes required

Reply - 
The contents have been revised from the readers perspective with the assistance of a native English speaker and both the contribution and significance of the research are emphasized as well.

Reply -
Add) Among the typical microphones, a dynamic microphone was selected for this study as standard equipment. Meanwhile, the method used in the process of a receiver transforming and representing the sounds into decibels was devised by borrowing an idea from Fleming’s Right-Hand Rule. In other words, the dynamic microphone operating as a receiver allows a sound wave (vibration) to be transformed into an electric signal through diaphragm vibration and we’ve used Fleming’s Right-Hand Rule as a calculation formula.

Comments and Suggestions for Authors
I would like to thank the Authors to have revised the paper. Unfortunately, the paper is still weak under the scientific point of view, but good in terms of the application presentation. In fact, many words have been spent for describing the problem, but very few for the implemented technical solutions. Moreover, nothing is written about the microphone equalization, neither about the microphones electrical characteristics.
It is also not clear how the receiver converts the sounds into the dB(A) values, mentioned in table 3.

Reply -
First of all, we are very grateful that you’ve read our research work again and given us an appropriate comment. Our response is being highlighted in red and we’ve attempted to make the thesis more meaningful by supplementing related work section to increase the level of contribution. Also, the additional works such as improving the pictures, explaining the algorithm more clearly, modifying the existing code with a pseudocode or deleting some of the contents have been performed to enhance the readability. Thus, we respectfully request your re-review if possible. The changes or additions made are also being highlighted in red.
The direction estimation algorithm is not mentioned at all, but only showing the results. This makes the description weak under the scientific point of view.

Reply -

Among the typical microphones, a dynamic microphone was selected for this study as standard equipment. Meanwhile, the method used in the process of a receiver transforming and representing the sounds into decibels was devised by borrowing an idea from Fleming’s Right-Hand Rule. In other words, the dynamic microphone operating as a receiver allows a sound wave (vibration) to be transformed into an electric signal through diaphragm vibration and we’ve used Fleming’s Right-Hand Rule as a calculation formula.

Add)
2.5. Direction Estimation Algorithm
The speed of sound (sound velocity) in the air is approx.. 340m/sec so that the time required for it to move 1cm is approx. 29us. However, as the minimum unit of measure that can detect the sound volume changes is ‘ms’, the sound waves cannot be measured. Thus, an algorithm which estimates the direction of sound based on vector computation was used.
The computation method of the algorithm using Table. 4 is as follows: Calculate individual average values from 3 sound measurements taken and align them in descending order to perform a vector computation using the highest and second highest values. This allows estimation of the latitude and longitude of a noise-generating location along with its direction based on the same of the current smartphone’s location. Meanwhile, Figure 2. describes a method of estimating the point of origin of the sound occurrence based on vector computation.

Figure 2. An example of direction estimation based on vector computation
In this situation, the position of ‘User’ was set as 0 on the distance measuring plane and then the proportion value was calculated by using of the highest and next highest noise values obtained from the two different locations [20-21]. Each proportion value can be calculated by adding those two noise values and then diving it with the respective noise values separately. Finally, by using the proportion values of the two locations, the point of origins of the noise in question is estimated to define it as a target location for investigation.
The distance between the current position and the target location and its angle are calculated by converting them into radian values. Since 1 latitude and longitude degree covers approx. 111km, their 7th (approx. 10cm unit) and 8th (approx. 1cm unit) values should be compared to perform an accurate calculation of an indoor position. The direction estimating algorithm using the radian values is shown in Table. 4.

Table. 4. The direction estimating algorithm
Step 1 Store the current position’s latitude and longitude when measuring the noise.
latitude = gps.getLatitude() longitude = gps.getLongitude()
Step 2 In getAveValue of ‘Comparison’, align individual decibel values in descending order and perform vector computation by using the highest and second highest values to estimate the point of origin of the noise.  

Estimated point of origin of Noise = [Direction angle of the highest noise]   ± 
[(Direction angle of the highest noise – Direction angle of the second-highest noise) * Proportion of two direction angles]
Step 3 Calculate the direction angles and distance between current position and estimated location in getDirection and getDistance.

  Distance = acos(sin(current position latitude) * sin(Estimated location Latitude)+cos(Current Position Latitude) * cos(Estimated Location Latitude)  * cos(Current Position Longitude – Estimated Location Longitude))
Direction= acos(sin(Estimated Location Latitude) sin(CurrentPosition Latitude) * cos(Distance))        /(cos(Current Position Latitude) * sin(Distance))) * (180/PI);

Step 4 Calculate direction angle and distance between the current position and estimated location.

2.6. Comparison with other system
The accuracy of the apps used for noise measurement was assessed in Research [22]. A smartphone (iPhone) was used and 37 apps from the 62 apps searched by using the keywords associated with ‘noise measurement’ were selected. From the measurement results, 31 apps exceeded the error range of ±2dB, which is an error range of common noise meters, whereas 18 apps exceeded the error range of ±5dB and the average error of 13 apps was ±11dB. It was necessary to pay special attention to most of these apps when using them as they did not seem to be completely fit for measuring noises. I the Research [23], the accuracy of the smartphone's noise measuring apps which can be used for the evaluation of noise generation in a working area was assessed to analyze their individual performance levels. As an experimental condition, the noise was gradually increased by 5dB at a time within the range of 60-95dB while using a pink noise having a frequency range of 20kHz. The result obtained by using a total of 20 iPhone apps and Android apps showed that they had an average error of 0.07dB when measuring noises within a particular range.
The existing study [22-23] had dealt with the accuracy of noise measuring apps commonly used for smartphones and found that many of them exceeded the error rate of actual noise meters or some of them produced higher accuracy only in a particular environment. Therefore, a step where the measurement values can be calibrated or compensated is required to improve when measuring noises with a microphone sensor embedded in smartphones.
Meanwhile, research [24] used a noise measuring method using the difference in the loudness and proposed an algorithm which enables a robot to estimate the location of the target. The mobile robot was able to generate the coordinates of a sound source with its sensor and estimated approximate bearing of the source by finding the direction while distinguishing 4 experimental areas distinguished by using a sound source tracking device.
The research [25] proposed a noise-tracking system using 4 microphones. By using a GCC-PHAT method, the system calculates the difference in the sound arrival times between the microphones and the optimal location was determined by using Iterative Least Square based on the arrival time differences. The accuracy of the measurements was in centimeters after comparing the actual location and the estimated location.
The research [26] proposed a method of locating the positions of microphone & speaker and a talker in an ad hoc network. A performance evaluation was carried out by assuming a situation where a participant sets his/her laptop computer on a conference table to communicate with others through the built-in microphone set. In this case, the distance error was 22cm between the actual and estimated positions of the microphone.
The research [27] proposed a method of determining the locations of microphone and sound source based on the difference in arrival times. The arrival times were calculated by arranging the flat-type microphone array in a grid form often referred to as a ‘unit cube’ to estimate locations.
  The research [28] proposed an algorithm which allows a mobile robot to track the position of a sound source by using 3 microphones. The angles of the sound entering into each microphone were measured for the estimation purpose. This method exhibited 3 ~ 10% higher accuracy than the existing Threshold-based method. There were many instances of performing performance evaluations in a limited environment in the existing research [24-28] as it estimated the direction by using specialized equipment such as microphone array, etc. However, such a method has a problem of applying it to an actual household so that another method which can be conveniently used at home with the common smartphones. Thus, this study proposes a power line communication (PLC)-based solution where noises are measured through the microphone sensor replacing CCTV to estimate and record/save the direction of the sound source and sound level (dB) as evidence.

Reviewer 2 Report

The changes have been processed correctly

Author Response

Comments and Suggestions for Authors
The changes have been processed correctly

Reply -
First, thank you for your detailed review. The contents have been revised from the readers perspective with the assistance of a native English speaker and both the contribution and significance of the research are emphasized as well. Thus, I’d like to respectfully request your re-review if possible. The contents added or changed are being highlighted in red.

Add 1) Among the typical microphones, a dynamic microphone was selected for this study as standard equipment. Meanwhile, the method used in the process of a receiver transforming and representing the sounds into decibels was devised by borrowing an idea from Fleming’s Right-Hand Rule. In other words, the dynamic microphone operating as a receiver allows a sound wave (vibration) to be transformed into an electric signal through diaphragm vibration and we’ve used Fleming’s Right-Hand Rule as a calculation formula.

Add 2)
2.5. Direction Estimation Algorithm
The speed of sound (sound velocity) in the air is approx.. 340m/sec so that the time required for it to move 1cm is approx. 29us. However, as the minimum unit of measure that can detect the sound volume changes is ‘ms’, the sound waves cannot be measured. Thus, an algorithm which estimates the direction of sound based on vector computation was used.
The computation method of the algorithm using Table. 4 is as follows: Calculate individual average values from 3 sound measurements taken and align them in descending order to perform a vector computation using the highest and second highest values. This allows estimation of the latitude and longitude of a noise-generating location along with its direction based on the same of the current smartphone’s location. Meanwhile, Figure 2. describes a method of estimating the point of origin of the sound occurrence based on vector computation.

Figure 2. An example of direction estimation based on vector computation
In this situation, the position of ‘User’ was set as 0 on the distance measuring plane and then the proportion value was calculated by using of the highest and next highest noise values obtained from the two different locations [20-21]. Each proportion value can be calculated by adding those two noise values and then diving it with the respective noise values separately. Finally, by using the proportion values of the two locations, the point of origins of the noise in question is estimated to define it as a target location for investigation.
The distance between the current position and the target location and its angle are calculated by converting them into radian values. Since 1 latitude and longitude degree covers approx. 111km, their 7th (approx. 10cm unit) and 8th (approx. 1cm unit) values should be compared to perform an accurate calculation of an indoor position. The direction estimating algorithm using the radian values is shown in Table. 4.

Table. 4. The direction estimating algorithm
Step 1 Store the current position’s latitude and longitude when measuring the noise.
latitude = gps.getLatitude() longitude = gps.getLongitude()
Step 2 In getAveValue of ‘Comparison’, align individual decibel values in descending order and perform vector computation by using the highest and second highest values to estimate the point of origin of the noise.  

Estimated point of origin of Noise = [Direction angle of the highest noise]   ± 
[(Direction angle of the highest noise – Direction angle of the second-highest noise) * Proportion of two direction angles]
Step 3 Calculate the direction angles and distance between current position and estimated location in getDirection and getDistance.

  Distance = acos(sin(current position latitude) * sin(Estimated location Latitude)+cos(Current Position Latitude) * cos(Estimated Location Latitude)  * cos(Current Position Longitude – Estimated Location Longitude))
Direction= acos(sin(Estimated Location Latitude) sin(CurrentPosition Latitude) * cos(Distance))        /(cos(Current Position Latitude) * sin(Distance))) * (180/PI);

Step 4 Calculate direction angle and distance between the current position and estimated location.

2.6. Comparison with other system
The accuracy of the apps used for noise measurement was assessed in Research [22]. A smartphone (iPhone) was used and 37 apps from the 62 apps searched by using the keywords associated with ‘noise measurement’ were selected. From the measurement results, 31 apps exceeded the error range of ±2dB, which is an error range of common noise meters, whereas 18 apps exceeded the error range of ±5dB and the average error of 13 apps was ±11dB. It was necessary to pay special attention to most of these apps when using them as they did not seem to be completely fit for measuring noises. I the Research [23], the accuracy of the smartphone's noise measuring apps which can be used for the evaluation of noise generation in a working area was assessed to analyze their individual performance levels. As an experimental condition, the noise was gradually increased by 5dB at a time within the range of 60-95dB while using a pink noise having a frequency range of 20kHz. The result obtained by using a total of 20 iPhone apps and Android apps showed that they had an average error of 0.07dB when measuring noises within a particular range.
The existing study [22-23] had dealt with the accuracy of noise measuring apps commonly used for smartphones and found that many of them exceeded the error rate of actual noise meters or some of them produced higher accuracy only in a particular environment. Therefore, a step where the measurement values can be calibrated or compensated is required to improve when measuring noises with a microphone sensor embedded in smartphones.
Meanwhile, research [24] used a noise measuring method using the difference in the loudness and proposed an algorithm which enables a robot to estimate the location of the target. The mobile robot was able to generate the coordinates of a sound source with its sensor and estimated approximate bearing of the source by finding the direction while distinguishing 4 experimental areas distinguished by using a sound source tracking device.
The research [25] proposed a noise-tracking system using 4 microphones. By using a GCC-PHAT method, the system calculates the difference in the sound arrival times between the microphones and the optimal location was determined by using Iterative Least Square based on the arrival time differences. The accuracy of the measurements was in centimeters after comparing the actual location and the estimated location.
The research [26] proposed a method of locating the positions of microphone & speaker and a talker in an ad hoc network. A performance evaluation was carried out by assuming a situation where a participant sets his/her laptop computer on a conference table to communicate with others through the built-in microphone set. In this case, the distance error was 22cm between the actual and estimated positions of the microphone.
The research [27] proposed a method of determining the locations of microphone and sound source based on the difference in arrival times. The arrival times were calculated by arranging the flat-type microphone array in a grid form often referred to as a ‘unit cube’ to estimate locations.
  The research [28] proposed an algorithm which allows a mobile robot to track the position of a sound source by using 3 microphones. The angles of the sound entering into each microphone were measured for the estimation purpose. This method exhibited 3 ~ 10% higher accuracy than the existing Threshold-based method. There were many instances of performing performance evaluations in a limited environment in the existing research [24-28] as it estimated the direction by using specialized equipment such as microphone array, etc. However, such a method has a problem of applying it to an actual household so that another method which can be conveniently used at home with the common smartphones. Thus, this study proposes a power line communication (PLC)-based solution where noises are measured through the microphone sensor replacing CCTV to estimate and record/save the direction of the sound source and sound level (dB) as evidence.
